# Is Dog Owner Obesity a Risk Factor for Canine Obesity? A “One-Health” Study on Human–Animal Interaction in a Region with a High Prevalence of Obesity

**DOI:** 10.3390/vetsci9050243

**Published:** 2022-05-22

**Authors:** Lourdes Suarez, Inmaculada Bautista-Castaño, Cristina Peña Romera, José Alberto Montoya-Alonso, Juan Alberto Corbera

**Affiliations:** Instituto Universitario de Investigaciones Biomédicas y Sanitarias, Universidad de Las Palmas de Gran Canaria, 35413 Las Palmas, Spain; lourdes.suarez@ulpgc.es (L.S.); inmaculada.bautista@ulpgc.es (I.B.-C.); cristina.pena.romera@gmail.com (C.P.R.); alberto.montoya@ulpgc.es (J.A.M.-A.)

**Keywords:** canine obesity, dogs, human–animal interaction, risk factors

## Abstract

Obesity in humans is a growing global problem and is one of the greatest public health challenges we face today. Most researchers agree that, as in humans, the incidence in the companion animal population is also increasing. The aim of this study was to evaluate the risk factors contributing to canine obesity in a region with a high rate of human obesity (Canary Islands, Spain), co-occurrence of obesogenic risk factors, and a canine population with a high percentage of unneutered dogs. We have focused on owner risk factors that promote obesity in humans, such as weight, lifestyle, nutritional habits, and low physical activity, among others. Thus, the human–animal interaction relationship that contributes to human obesity and influences canine obesity has been studied. A multicentre cross-sectional analytical study of 198 pairs of dogs from urban households and their owners was used. A multivariable logistic regression study was completed to analyse owner characteristics variables associated with canine obesity. This transdisciplinary study was conducted with physicians and veterinarians using a “One Health” approach. Our results suggest that, in a region of high obesogenic risk, obese/overweight dogs are primarily female, older than 6 years, and neutered. Being an overweight dog owner was found to be the most important factor in the occurrence of obesity in dogs. Owners of overweight dogs were mainly females, older than 40 years, who did not engage in any physical activity. A strong correlation has been found between dog owners with low levels of education and obesity in their dogs. We suggest that veterinarians should develop and design strategies to encourage pet owners to engage in physical activity with their dogs for the benefit of both.

## 1. Introduction

### 1.1. Human Obesity

Obesity is defined as an accumulation of excessive amounts of adipose tissue in the body and the most common nutritional disorder in pets [1]. Obesity is usually the result of either excessive dietary intake or inadequate energy utilisation, which causes a state of positive energy balance [2]. Obesity is an escalating global problem in humans and is one of the greatest health challenges faced today [3]. Being overweight is an increasingly common problem in both humans and their pets; this is particularly true in developing countries, and especially in urban settings [4].

Population data available in Spain show a prevalence of obesity (BMI ≥ 30 kg/m^2^) of 14.5% in adults aged 25 to 60 years; estimates are based on individual measurement of body weight and height [5,6]. The geographical distribution of the problem shows a trend with higher rates in the south-southeast part of Spain, including Andalucía, Murcia, and the Canary Islands [5].

The Canary Islands is one of the Spanish Regions with the highest prevalence of human obesity [7]. Although there was a slight decrease in the prevalence of obesity in 2015 compared to 2009 (from 19.54 to 18.64%) [8], the latest study of the prevalence of obesity and their associated cardiovascular risks showed that 23.1% of the Canarian population was obese and 35.1% was overweight [7].

In humans, the medical importance of obesity lies in its effect on the mortality and morbidity of associated diseases [9]. Similarly, obesity promotes the development of a number of diseases that reduce the quality of life and longevity of dogs [10,11,12].

### 1.2. Dog Obesity

Obesity in pets is a growing concern, and its increasing incidence appears to be mirroring the trend observed in humans. The main reason for the development of obesity in dogs is an imbalance in the energy balance equation. In this respect, either excessive dietary intake or inadequate energy utilisation can lead to a state of positive energy balance, resulting in excess body fat storage [13].

Most researchers agree that, as in humans, the incidence of obesity in the pet population is increasing and leads to decreased life expectancy. Obesity is associated with cardio-respiratory, orthopaedic, endocrine, and oncologic disorders in both humans and dogs [14,15,16,17,18,19]. The Global Pet Obesity Initiative Position Statement (GPOIPS) was proposed by the Association of Pet Obesity Prevention (APOP), a non-profit organisation founded in 2005 and committed to making the lives of dogs, cats, all other animals, and people healthier and more vital [20].

Data on the incidence of this problem in dogs vary according to different authors. Studies in recent years have shown that the prevalence of canine obesity varies between 25 and 44% in developed countries [21,22,23]. The prevalence of obesity in dogs is increasing [24]. In 2014, obesity was the seventh most prevalent disorder (6.1%, 95% CI 5.2–7.1) reported in dogs in UK primary veterinary clinics [25]. In 2021, obesity became the third most prevalent disorder (7.07%, 95% CI 6.74–7.42), after periodontal disease (12.52%) and otitis externa (7.30%) [26]. Moreover, according to the Pet Obesity Survey results published in 2018 by APOP, the prevalence is increasing. The most recent studies of prevalence estimated that the 59.5% of cats and 55.8% of dogs in the United States were overweight or obese. In addition, new standards have been established for dogs, defining obesity as a weight greater than 15% of the ideal weight [27,28,29]. Therefore, APOP recommends the use of the term “above the ideal weight”, because it allows veterinarians to use other strategies to identify the onset of obesity in addition to the BCS definition [20].

Cultural influences are likely to affect not only pet owners, but also their dogs. For example, a study comparing overweight and obesity in Thailand and the Netherlands concluded that, in warm countries, pet owners are less likely to exercise with their dogs. These cultural influences could negatively impact the dog’s quality of life [13].

Since pets and their owners are not genetically related, and pets depend, in terms of their food and lifestyle, on the attitudes and behaviour of their owners, the relationship between the weight of owners and that of their pets provides information on the effect of individual attitudes and behaviour on body weight [30].

Several studies have shown that both the weight and health of owners influence the eating habits and physical activity of their dogs, as well as the presence of canine obesity and associated diseases. In 1970, Mason demonstrated a relationship between obesity in pet dogs and the degree of obesity in their owners [31]. More recently, this finding was confirmed by Bland et al. [32] and Nijland et al. [30].

### 1.3. Human–Animal Interaction

The benefits of human–animal relationships for human health and well-being have attracted scientific interest for years [33,34,35,36]. This is not only evidenced in the veterinary research literature [16,37], but also in human medicine publications [38,39], including the human paediatric literature [40,41,42,43,44].

In humans, the perception of health and illness, and, hence, quality of life, is culturally determined [37,45], and thereby, influence the relative importance of factors, such as affection, satisfaction, and acceptance. In turn, these factors are influenced by lifestyle, past experience, and social consensus [46]. A person’s beliefs and attitudes towards being overweight, obesity, the eating behaviour, and exercise are also culturally determined [47]. For example, many cultures believe that being overweight reflects wealth, prosperity, and high status [45,47]. However, it is not clear whether cultural beliefs influence how owners perceive the quality of life of their dogs [13,37].

A very interesting study on obesity prevention in youth with autism spectrum disorder emphasises the introduction of nutrition-related ideas through the lens of dog health. Emphasising how overeating and obesity can lead to disease in dogs without inducing guilt or reproach in children is a good example of how pets can help to promote a healthy lifestyle for humans [48].

Inversely, a growing interest in the human influence on companion animal health, especially obesity in dogs, has been evidenced by an increase in publications related to this topic [19,49]. Ronja and Kölle [50] have reviewed the most important factors contributing to obesity in dogs, a disease that is often unrecognised by pet owners [51,52]. This is a particular challenging task for veterinarians in general practice when talking to owners about overweight and its comorbidities [53,54,55].

There are many studies in the available literature related to the association of owners’ beliefs and behaviour with obesity in pet dogs [52,56,57]. Recently, Webb et al. [37] investigated theoretical frameworks for identifying and measuring factors that are potentially associated with obesity among dogs. Additionally, factors that contribute to improving the effectiveness of a weight loss plan for companion animals has been analysed from a “One Health” approach [58,59].

Further studies are needed to understand how owners’ behaviour negatively influences pet health, especially if they provide neglectful care. The results of Coy et al. [60] suggest that attachment plays an important role in shaping the relationship between pet and caregiver and influences several elements that contribute to pet obesity, especially in dogs. Many questions remain unanswered about health and behaviour conditions affecting dogs, especially those influenced by multiple factors. The “Generation Pup” study [61] has been recently launched which, through a longitudinal study in purebred and mixed breed puppies, will provide insight into the environmental and genetic factors affecting various disorders, including obesity. The results of that study will be known in the next few years.

Thus, our study was aimed to describe the influence of dog ownership on canine obesity and bring a new perspective to strategies to reduce pet obesity and pet health through a “One Health” approach.

## 2. Materials and Methods

For the purposed aim, a transdisciplinary study was conducted with the participation of physicians and veterinarians. Factors related to age, sex, weight, lifestyle, and physical activity of owners that contribute to and influence canine obesity were evaluated in a group of dogs in the Canary Islands (Spain), a region with a high rate of human obesity and co-occurrence of obesogenic risk factors. To obtain enough data for an appropriate statistical analysis, it was needed to complete this study over a two-year period. Our study is an example of a collaborative, multisectoral, and transdisciplinary approach with the aim of achieving optimal health outcomes recognising the interconnectedness between humans and animals and their shared environment in line with the “One Health” approach [62,63].

### 2.1. Animals

The study was an analytical cross-sectional multicentre study of a convenience sample consisting of 198 pairs of dogs from urban households and their owners who visited several veterinary clinics for routine consultations on the island of Gran Canaria (Canary Islands, Spain). The study protocol was approved by the Ethics Committee of the Veterinary Medicine Service of the University Foundation of Las Palmas (Certificate number MV-2009/01).

A complete clinical examination, including a complete blood analysis, was performed in all the animals. Dogs diagnosed with chronic diseases, such as chronic liver disease, kidney disease, diarrhoea, vomiting, endocrine disorders, and long-term pathological anorexia, were not included in the study.

The weight status of the dogs was determined by veterinarians using the Body Condition Score (BCS) and assessed according to a nine-point scale [64]. Dogs were considered to have excess weight (overweight and obese) when the BCS was 6 or higher. Dogs were considered to have normal weight when the BCS was 4 or 5.

The study sample was divided into two groups according to the weight of the dogs (overweight group and normal weight group). The sample of the overweight group consisted of a group of 137 pairs of dogs with excess weight (overweight and obese) and their owners. The sample of normal weight group sample (negative group) consisted of a group of 61 pairs of normal weight dogs and their owners.

### 2.2. Dog Owners

All owners gave written consent to participate in this study. The dog owners were studied in the Canarian Institute of Medicine and Nutrition (ICAMEN) by a physician and specialist in obesity and human nutrition. The study was conducted in accordance with deontological standards and the European legislation on animal protection.

Owners were asked to fill a questionnaire with questions on characteristics, such as age and sex, level of education, absence of physical activity (referring to owners), smoking, alcohol consumption, and employment status. The level of education completed was divided into three levels: low level, no formal education or education at the lowest level of secondary school; medium level, education at the middle level for the sample; and high level, education at a level of at least advanced professional or university education (for the comparison in the statistical analysis, these last two levels were grouped together).

The weight status of the owners was determined by Body Mass Index (BMI), calculated as weight/height^2^ (in kg/m^2^). Weight was measured on a Roman scale, SECA^®^ model 712 with a capacity of 200 kg, in 100 g increments. Patients were lightly clothed and without shoes. Height was measured at the same outpatient visit using a SECA 221 measuring scale with a range of 6–230 cm, in 1 mm divisions. For statistical purposes, the patients were divided into the following groups and based on the “Clinical Guidelines on the Identification, Evaluation and Treatment of Overweight and Obesity in Adults” of the USA Expert Committee on Obesity [65]. These groups were as follows: normal weight: BMI < 25 kg/m^2^; overweight (overweight and obese): BMI > 25 kg/m^2^.

### 2.3. Statistical Analysis

For the statistical analysis of the data, the statistical package SPSS version 27.00 for Windows (SPSS Inc., Chicago, IL, USA) was used. In all cases, the level of significance was *p* < 0.05. The descriptive analysis of the variables was carried out by studying proportions in the case of qualitative variables and using means of central tendency (mean or median) and measures of dispersion (standard deviation) for quantitative variables. Kolmogorov–Smirnov test was applied to the continuous variables to assess their normal distribution.

In the case of continuous variables with Gaussian distribution, the comparison of absolute means between two groups was performed using Student’s *t*-test, and for the comparison of absolute means between three or more groups, analysis of variance (ANOVA) was used.

For continuous variables with asymmetric distribution, non-parametric tests were used: the Wilcoxon rank sum test for the comparison of absolute means between two groups and the Kruskall–Wallis test for the comparison of absolute means between three or more groups.

A bivariate analysis of proportionality of the distribution of categorical variables was estimated using the χ^2^ test and, when required, Fisher’s exact test was used. Linear trend was estimated using the Mantel–Haenszel linear trend test.

For the study of the association between canine obesity and various factors, multivariable logistic regression analysis was performed. Odds ratio (OR) and the confidence interval (CI) at 95% confidence of the OR value was studied. Goodness-of-fit was assessed using the Hosmer–Lemeshow test.

## 3. Results

### 3.1. Animals

Of the 198 dogs examined, 137 (69.2%) were classified as overweight/obese. Then, 61 (30.8%) were normal weight dogs. The mean age of all dogs was 6.51 ± 3.69 years. Table 1 shows the distribution of the dogs according to the different variables studied: BCS, sex, age, and reproductive status.

In the group of overweight/obese dogs, 98 belong to 25 recognised breeds, mostly Yorkshire Terriers, German Shepherds, miniature poodles, Cocker Spaniels, bulldogs, boxers, and Canarian Dogos, and 39 were mixed breeds. All were considered adult for their breed; age ranged from 2 to 14.7 years. The average age was 6.77 ± 3.36 years.

In total, 50 dogs were male (36.49%), of which 96% were intact, and 87 dogs were female (63.50%), of which 85% were intact. We found statistical differences (*p* = 0.042) according to sex in unneutered dogs. Specifically, we found a higher prevalence of obesity in the unneutered females (73.3%) compared with unneutered males (58.4%). In the group of normal weight dogs, 41 were of recognised breeds, which was similar to the group of overweight/obese dogs, and 20 were mixed breeds. All were considered adult for their breed, and age ranged from 2 to 15.1 years. The average age was 6.87 ± 2.96 years; 23 dogs were male (37.70%), of which 91.3% were intact, and 38 dogs were female (62.29%), of which 84.21% were intact.

### 3.2. Dog Owners

On the other hand, dog owners aged between 18 and 84 years, and 83 (41.9%) were men and 115 (58.1%) were women. Table 2 shows the distribution of owners of both overweight/obese and normal weight dogs according to several characteristics, including the weight status and sex of the owner, among others. In the group of owners of overweight/obese dogs, 40.1% (55/137) were men and 59.9% (82/137) were women; 33.6% (47/137) of owners were in the normal weight category, whereas 66.4% (91/137) of them were overweight/obese (BMI ≥ 25 kg/m^2^). The sex breakdown for this latter subgroup of overweight/obese owners having dogs with overweight was 44% (40/91) men and 56% (51/91) women.

Regarding the owners of dogs with normal weight, 62.3% (38/61) were also in the owner normal weight category, whereas 37.7% of them (23/61) were found to have overweight (BMI ≥ 25 kg/m^2^); 59.1% (13/22) of this latter subgroup (i.e., overweight/obese owners having normal weight dogs) were women and 40.9% (9/22) were men. Other characteristics of the owners are shown in Table 2.

### 3.3. Human–Animal Interaction

The owners of overweight/obese dogs were significantly older than those owning normal weight dogs (*p* = 0.016). Additionally, we have found significant differences when comparing the owners of overweight/obese dogs having only basic education or no studies and those with an average (secondary) to high level (university) of education (*p* = 0.002).

The majority of owners of dogs with overweight 62.88% (86/137) were found to have a lack of physical exercise (*p* = 0.02). No significant differences in smoking habits and alcohol intake were found between the owners of overweight/obese dogs and the owners of dogs with normal weight.

Non-statistical differences have been found comparing the work situation of owners and the obesity of dogs (*p* = 0.152). However, we want to highlight that the majority of owners of overweight/obese dogs who were unemployed also tended to have overweight themselves, 81.6% (31/38), and the majority of them, 71.0% (27/38), were women. In total, 100% (61/61) of owners of non-obese dogs considered their dog to be healthy; however, 86.1% (118/137) of owners of overweight/obese dogs also considered that their dogs were healthy (*p* = 0.001).

The logistic regression analysis (Table 3) shows that dogs with overweight/obese owners are three times more likely to be overweight/obese than those whose owners were of a normal weight (*p* = 0.001). Likewise, dogs with owners who did not exercise were two times more likely to be overweight/obese (*p* = 0.020).

## 4. Discussion

### 4.1. Animals

Demographic risk factors for the development of overweight status in dogs has been previously reported and include being middle-aged, female, and neutered [66,67], which is in line with our overall results detailed in Table 1. However, there is an increase with age in the proportion of dogs that had been neutered, so it is difficult to separate the effect of age from the effect of neutering [66]. Our global results regarding the effect of neutering in the risk of canine obesity do not differ from those published widely in the literature. Then, the effect of neutering as a risk of obesity is reduced or limited in our results. The high percentage of unneutered animals included in our survey (84.3%; 167/198) allow a better understanding of the contribution of the attitudes and behaviour of owners as a risk factor for canine obesity. We have found statistical differences (*p* = 0.042) in the comparison of the prevalence of obesity of intact females, 58.4% (45/77), with the intact males, 73.3% (66/90). In 2005, McGreevy et al. [66] did not show a relationship between obesity and the sex of the dogs. Our results also agree with those published by Mao et al. [29], which found a higher percentage of obesity in intact males (47.3%) compared with females (38.7%). Pegram et al. [22] found similar risk estimates in female neutered dogs (OR = 1.89) and male neutered (OR = 1.90), whilst male intact dogs had 1.23 times the odds (95% CI 1.04 to 1.46) of overweight status compared with intact female dogs. Moreover, a later study published by Bjørnvad et al. [68] evidenced that neutering dramatically increased both BCS and the risk of being heavy/obese in male dogs, but not in bitches. This is particularly interesting when other factors, such as the attitudes and behaviour of owners, as described by Bland et al. [32] may contribute to the risk of obesity. On the other hand, neutering cats has been shown to increase the risk of obesity, but long-term studies have not shown a correlation between age at neutering and risk of obesity [69]. Thus, future studies are needed to clarify the contribution of the dog sex and neuter status to the risk of obesity in different geographical and socio-cultural societies around the world.

### 4.2. Dog Owners

In our study, dog owner overweight was found to be the most important factor in the occurrence of obesity in dogs. Dogs of overweight owners were >3 times more likely to be obese (OR = 3.05; 95% CI 1.57–5.82; *p* = 0.001). Moreover, we found a positive relationship between the degree of obesity of dogs and the BMI of their owners; 66.4% of the owners of overweight/obese dogs were also found to have overweight. The positive relationship between the weight of companion animals and that of their owners has been previously described [2,30,49,51]. However, further studies are needed to understand the relationship between owner obesity and dog obesity and to determine the exact mechanism underlying the obesity association between them.

In agreement with van Herwijnen et al. [70], our results indicate that owners may apply their personal attitudes and behaviour to their pets. In obese people, this behaviour has been referred to as an obesogenic lifestyle “eating in excess and inactivity” [70]. Additionally, sex differences in human obesity have been reported and have been associated with sedentary lifestyle. Another study by Hermann et al. [71] also confirms that women with a university degree had a 2.1 kg/m^2^ lower BMI as compared to women with the lowest level of education. The owner shares this obesogenic lifestyle with their pets, which could be considered an indicator of over-humanising. The correlation between dog obesity and human obesity from a “One Health” approach has recently become the subject of research [49].

### 4.3. Human–Animal Interaction

Dog obesity is influenced by their owner’s attitudes, and it is quite possible that those same attitudes influence an owner´s health status. This could explain why in both our study and in others, there is a correlation between obesity in owners and their dogs [30,49].

Muñoz-Prieto et al. [51] clearly showed that obesity in dogs is affected by the interrelationships between food management, exercise, and social factors. These are causes shared by man and dog and give rise to the high incidence of this process in man in the Canary Islands (Spain), which, as mentioned above, is an obesogenic risk area [8,72].

The identification of animal overweight by dog owners is a major concern for veterinary practitioners, animal health, and nutrition companies [32,54,55,59,73]. Our study also reveals that obese owners with obese dogs consider their dogs to be healthy (86%). Another study conducted in 10 European countries also revealed that dog owners did not perceive obesity in dogs to be a disease [51]. Therefore, the belief that obesity is not a health problem may negatively impact the quality of dog’s life. On the other hand, in the social media and the animal welfare literature, we have found a current discussion regarding the potential of prosecuting dog owners for animal cruelty because of allowing animal obesity to progress to extremes. Some countries have started to view overfeeding/underexercising as a form of cruelty. The APOP has echoed these social discussions [20].

In addition, our results indicate that women are more likely to have overweight/obese dogs than men, especially if owners do not exercise. Abdominal obesity is more common in Spanish men than in women [6]. However, a sedentary lifestyle is more common in women than in men in Spain [74]. Differences in sex fat distribution and in eating behaviour may also play a key role in the fat distribution [6].

Interestingly, obesity levels in dogs did not differ between dog owners who smoke (*p* = 0.522) and drink alcohol (*p* = 0.226) and those who do not. It is important to point out that 73.91% (17/23) of the owners with overweight/obese dogs and a low level of education or no formal education were also overweight/obese, and 59.9% (82/137) were women. This latter finding is in line with the study conducted by Perez-Rodrigo et al. [5], which shows that obesity rates are higher among women aged 45 years or over, of a lower social class, and living in semi-urban areas.

A strong correlation between pet owners with low levels of education and obesity in their dogs has been found in our study (*p =* 0.002). However, according to the logistic regression model used, the calculated probability of dog’s obesity when owner’s education levels are low was not significant (*p* = 0.535). Therefore, we cannot estimate the risk of obesity in the dog when the owner’s education levels are low. Similarly, human obesity is 5% more prevalent in the lowest level of education than more educated subpopulations in Spain [75]. In many developed countries with high incomes, obesity is inversely associated with the level of education. Our findings agree with those of Marques-Vidal et al. [76], who found a greater prevalence of obesity in the low-education strata of the population.

In the present study, we have found that the majority of overweight owners of dogs were women, 55.2% (63/114), albeit with no significant differences in comparison with male owners. A recent study found that the variables related to an increased body fat index (BFI) and Body Condition Score (BCS) in the dog were the sex of the owner (men were more susceptible than women) and also whether the owners had unhealthy habits (for example, smoking) [51]. These data contrast with those found in our study, where the sex of owners of obese dogs and the influence of healthy habits on the dog’s BCS was not significant.

Regarding the age of the owners, several scientific studies have shown that the age of the owner is a risk factor for dog obesity. Muñoz-Prieto et al. [51] calculated that risk in the European population (OR 1.1, 95% CI 0.65–1.21; *p* = 0.046). Thus, the dogs of older owners are more likely to be obese. We found a similar finding in our study where most owners of overweight dogs were 40 years old or older (*p* = 0.016). However, according to the logistic regression model used, the calculated probability of dog’s obesity when owners were 40 years old or older was not significant (*p* = 0.274). Therefore, we cannot estimate the risk of obesity in the dog when owners were 40 years old or older. Endenburg et al. [13] have described owners of obese dogs as having less interest in dog nutrition than owners of normal weight dogs. Similarly, we have found that almost 25% of dog owners with overweight were in the low-level education bracket or had no formal education; 73.9% of these people were also obese and the majority were women.

According to Endenburg et al. [13], owners of obese dogs do not engage in physical activity with their dogs. Owning a dog has been linked to increased physical activity, decreased obesity rates, overall health benefits and a lower mortality rate than the general population [77,78,79]. However, in our population, the percentage of dog owners who were overweight/obese was quite high (66.4%), indicating that having a dog does not necessarily protect people from being overweight. We found that most dog owners with overweight (62.77%) did not engage in physical exercise and the majority were women. Numerous factors may influence the relative ease with which weight is gained by dogs, and these include genetics, age, reproductive status, the amount of physical activity, and the calorific content of the diet. Other recognised associations in dogs include an indoor lifestyle, inactivity, and middle age [13].

The owners’ perception of obesity in themselves and their dogs also requires further investigation in the search for potential ways to improve the prevention and control of obesity. However, good communication between owners and veterinarians is important for adherence to diet and physical activity recommendations [1], and any obesity preventive measures must be targeted according to the owner’s educational level [76]. With the combination of a calorie-restricted diet and increased physical activity by exercising together, it is possible for both pet owners and their dogs to lose weight [51,80,81,82].

### 4.4. Limitations

The average prevalence of human obesity in Spain can be somewhere in the middle between the northern European countries, with low proportions of obesity, and the USA and eastern European countries, with higher rates. However, the Canary Islands (Spain) must be included in the areas with a higher proportion of obesity (higher than 20%) [7], and, therefore, it is considered an obesogenic risk zone [8,72]. Therefore, the results of the present study may be biased due to the special characteristics of the population studied.

Another limitation is the possible underestimation of overweight status by veterinary professionals, which is well-recognised in the literature [83,84]. However, the criteria for the definition of overweight and obesity status in dogs based on BCS were explained in detail to all the veterinarians before their participation in this study to reduce reporter bias. On the other hand, all the owners were evaluated by the same physicians and the calculation of the BMI scores results in more precise data with low reporter bias. Another limitation of our study is concerned with the perception of people about obesity and overweight of their pets. This limitation has been reported in the literature [51,52].

## 5. Conclusions

In conclusion, our results suggest that, in a region of high obesogenic risk (23% and 35% of the human population with obesity and overweight, respectively), obese/overweight dogs are mainly males, older than 6 years old, and neutered. We found a higher prevalence of obesity in unneutered males compared to unneutered females. These differences were significant between the sexes in unneutered dogs.

Regarding the human–animal interaction, we found that obese/overweight dogs were more prevalent in obese/overweight owners. Dog owner overweight was found to be the most important risk factor for the occurrence of obesity in dogs. Dogs with overweight owners (men and women) were more likely to be overweight. However, we observed that obesity in dogs was more prevalent when their owners were overweight women older than 40 years with little education who did not engage in physical activity. 

We suggest that veterinarians and physicians from a “One Health” approach must develop and design strategies to encourage pet owners to engage in physical activity with their dogs for the benefit of both.

## Figures and Tables

**Table 1 vetsci-09-00243-t001:** Distribution of the dogs according to BCS, sex, age, and reproductive status.

Variables	Categories	Normal Dogs = 61	Overweight/Obese Dogs = 137	Total (198)	*p*-Value
BCS	4	15 (7.6%)	-	61 (30.8%)	-
	5	46 (23.2%)	-		
	6	-	28 (14.1%)	101 (388%)	
	7	-	45 (22.7%)		
	8	-	46 (23.2%)	64 (32.3%)	
	9	-	18 (9.1%)		
Sex	Females	28 (24.3%)	**87 (75.7%) ***	115 (58.1%)	**0.016**
	Males	33 (39.8%)	50 (60.2%)	83 (41.9%)	
Age	<6 years old	42 (43.3%)	55 (56.7%)	97 (48.9%)	**0.001**
	>6 years old	19 (18.8%)	**82 (81.2%) ***	101 (51.1%)	
Neutering	Not neutered	56 (33.5%)	111 (66.5%)	167 (84.3%)	**0.039**
	Neutered	5 (16.1%)	**26 (83.9%) ***	31 (15.7%)	

* Statistical differences (*p* < 0.05) are related to the data marked with an asterisk. Statistical differences are highlighted in bold.

**Table 2 vetsci-09-00243-t002:** Results of the studied variables related to the characteristics of the owners. The distribution of categorical variables was estimated using the χ^2^ test and, when required, Fisher’s exact test was used. Statistical differences are highlighted *.

Owner Characteristics		Normal Dogs = 61	Overweight/Obese Dogs = 137	Total	*p*-Value
BMI (>25 kg/m^2^)	Overweight/obese	**23 (20.2%)**	**91 (79.8%)**	114 (57.6%)	**<0.001**
	Normal weight	38 (45.2%)	46 (54.8%)	84 (42.4%)	
Sex	Women	28 (33.7%)	55 (66.3%)	83 (41.9%)	0.273
	Men	33 (28.7%)	82 (71.3%)	115 (58.1%)	
Age	<40 years old	37 (38.5%)	59 (61.5%)	96 (48.5%)	**0.016**
	>40 years old	**24 (23.5%)**	**78 (76.5%)**	104 (51.5%)	
Habitual smoker	Yes	24 (30.4%)	55 (69.6%)	79 (39.9%)	0.522
	No	37 (31.1%)	82 (68.9%)	119 (60.1%)	
Alcohol intake (2–3 drinks/day)	Yes	2 (16.5%)	10 (83.5%)	12 (6.1%)	0.226
	No	59 (31.7%)	127 (68.3%)	186 (93.9%)	
Level of education	Non or Basic education	**3 (11.5%)**	**23 (88.5%)**	26 (13.1%)	**0.002**
	Secondary or University education	58 (33.7%)	114 (66.3%)	172 (86.9%)	
Work situation	Employed	49 (33.1%)	99 (66.9%)	148 (74.7%)	0.152
	Unemployed	12 (24.0%)	38 (76.0%)	50 (25.3%)	
Physical activity	Yes	28 (35.4%)	51 (64.6%)	79 (39.9%)	**0.020**
	No	**33 (27.7%)**	**86 (72.3%)**	119 (60.1%)	
Do you think your dog is healthy?	Yes	**61 (34.1%)**	**118 (65.9%)**	179 (90.4%)	**0.001**
	No	0 (0%)	19 (100%)	19 (9.6%)	

* Variable results with statistically significant differences between the expected frequencies and the observed frequencies are marked in bold.

**Table 3 vetsci-09-00243-t003:** Results of the logistic regression analysis completed for the study of the association between the overweight/obesity of dogs and the different owner variables’ categories studied.

Owner Variables (Category)	B	SE	Wald	Df	*p*-Value	Odds Ratio	95% CI	
							Lower	Upper
Overweight (BMI ≥ 25 kg/m^2^)	1.106	0.334	10.966	1	**0.001**	**3.023**	1.571	5.819
Owner age (>40 years old)	0.397	0.364	1.194	1	0.274	1.488	0.729	3.035
Employment status (Unemployed)	0.262	0.417	0.395	1	0.530	1.300	0.574	2.946
Owner sex (Women)	−0.252	0.366	0.472	1	0.492	0.777	0.379	1.594
Education of owners (non or basic studies)	0.189	0.305	0.385	1	0.535	1.208	0.665	2.196
Smoker (Yes)	0.234	0.356	0.433	1	0.510	1.264	0.629	2.540
Frequency of physical activity (No)	1.278	0.262	9.935	1	**0.020**	**2.758**	1.552	5.041
Constant	−0.099	0.826	0.014	1	0.905	0.906		

B—Logistic Regression Coefficient, SE—Standard Error, Wald—Wald Chi-Squared, Df—Degrees of freedom, CI—Confidence Interval. Statistical differences are highlighted in bold.

## Data Availability

The data presented in this study are available on request from the corresponding author. The data are not publicly available due to no public funding.

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
