# Peer review of "Is Dog Owner Obesity a Risk Factor for Canine Obesity? A “One-Health” Study on Human–Animal Interaction in a Region with a High Prevalence of Obesity"

_vetsci, 2022, doi:10.3390/vetsci9050243_

Round 1

Reviewer 1 Report

Authors aimed to describe the influence of dog ownership on canine obesity and bring a new perspective for strategies to reduce pet obesity and pet health through a “One Health” approach.

Specific comments/suggestions are highlighted in the attached .pdf.

Some point in the discussion refer to univariate evaluations, when the multivariate model is indicating that they are not significant to determine obesity in dogs, it is important to consider this point.

Also it is recurrent to read in the manuscript the quote "data not shown on Tables", if it is so recurring to indicate that, it may be necessary to include that data in it.

Author Response

Dear Reviewer 1.

Thank you very much for your suggestions. Now the paper has been substantially improved. We really appreciate your support!

Here and attached in PDF are our answer to your suggestions.

Responses to your suggestions:

We have corrected all the spelling mistakes. Thanks for the revision!

Line 134.

Really, we think that these sentences are more appropriate for the end of the Introduction. However, they can be moved to the beginning of the Material and Methods, just before the detailed descriptions. So, these sentences have been moved to the beginning of Material and Methods. If you consider another place more appropriate, we really appreciate your suggestions.

Related to the sentences “Data not shown in tables.” Has been included 6 times:

- Line 222

This sentence was included by a proposal of a previous reviewer. Here the combination of two variables results is shown within the text and not expressed in Table 1. In that table, a combination of different variables cannot be included, and this sentence was added to specify the readers not to look at the table for these results.

So, “Data not shown in tables” could be eliminated and readers could easily understand these results. No more information is required.

We found statistical differences (P = 0.042) according to sex in unneutered dogs. Specifically, we found a higher prevalence of obesity in the unneutered females (73.3%) compared with unneutered males (58.4%).

- Lines 238, 247, 285, 357.

Like line 222, data results are detailed in the text. So, “Data not shown in tables” could be eliminated also.

- Line 261

In order to introduce the percentage and the total of animals to avoid the sentences “Danta not shown in tables”, we have revised again our results from the raw data, really the results were not clear for us when reading again the percentage presented previously. Then we have found a mistake. Probably, when writing the paper, we turned the results of intact males to intact females and vice versa. So, we have checked these results twice and now the presented results are correct.

Then the discussion sentences related to a disagree with previous results has been changed to “agree”. And the final sentences written in the conclusion related to the profile of the overweight dogs has been changed to males instead of females (previous line 409).

We are very sorry for this mistake! This could be attributed to the huge amount of data that we have managed in our data sets and a mistake made when transcribing the data into word. Sorry!

Then the sentences is:

We have found statistical differences (P = 0.042) in the comparison of the prevalence of obesity of intact females 58.4% (45/77) (73.3%) with the intact males 73.3% (66/90)(58.4%)

Line 269. Table 3

Owner variables in the first column has been changed for the categories related to the risk calculated. Thanks, very good idea! Clearer to the readers now!

- Line 351 and 367.

Related to the multivariable model and the significance. You are right! 

The existence of a significant relationship between a qualitative variable with two levels and a continuous variable can be studied using statistical tests such as t-test or ANOVA. However, logistic regression also allows us to calculate the probability that the dependent variable belongs to each of the two categories depending on the value of the independent variable.

Then, we have added a sentence for clarifying this to the readers.

Line 351 and followings

A strong correlation between pet owners with low levels of education and obesity in 350 their dogs has been found in our study (P=0.002). However, according to the logistic regression model used, the calculated probability of dog’s obesity when owner’s education levels are low was not significant (P = 0.535). Therefore, we cannot estimate the risk of obesity in the dog when the owner's education levels are low. Similarly, human obesity……...

Line 367 and followings

So, the dogs of older owners are more likely to be obese. We found similar finding. In our study most owners of over-weight dogs were 40 years old or older (P=0.016). However, according to the logistic regression model used, the calculated probability of dog’s obesity when owners were 40 years old or older was not significant (P = 0.274). Therefore, we cannot estimate the risk of obesity in the dog when owners were 40 years old or older. Endenburg et al. [13] have described ……….

Reviewer 2 Report

Major Comments:

Thank you for your responses to the reviewers' suggestions. The reader should still be able to understand why 198 total dogs were chosen for study. As it is written now, that still is not clear. Was a certain time frame selected, or similar? For the p-values in Table 2, it will be helpful for the reader if you clarify what test was used to obtain that p-value.

Line 419 - I am not a veterinarian, but please check to determine the appropriateness of a veterinarian recommending physical activity to pet owners "for the benefit of both." I assume that it is not appropriate for vets to make recommendations regarding human health.

Minor Comments:

  • Line 215 - Only the proper names (Yorkshire, German, Canarian, etc.) should be capitalized. Bulldogs/Boxers/Poodles should not be capitalized.
  • The number of decimal places (one or two) is not consistent when presenting percentages. I suggest picking one or the other for consistency. The same goes for decimal places for confidence intervals (ex. Line 303).
  • Line 315 - use the abbreviation only here (BMI had already been defined). The same goes for APOP in Line 337. (Also there are two periods in this sentence).
  • Line 339 - clarify that you mean especially if the women (not the dogs) do not exercise, if this is your intent.
  • Line 416 - overweight dogs were not more prevalent in overweight women - obesity in dogs was more prevalent when their owners were overweight women older than 40 years with little education who did not engage in physical activity.

Author Response

Answer to Reviewer 2

Thank you very much for your suggestions. Now the paper has been substantially improved. We really appreciate your support!

Here and attached you will find our answer to your suggestions:

Responses to your suggestions:

We have corrected all the spelling mistakes. Thanks for the revision!

The reader should still be able to understand why 198 total dogs were chosen for study. As it is written now, that still is not clear. Was a certain time frame selected, or similar?

The owners’ participation on the study was voluntary and required not only a visit to the veterinary clinics (where one of our team members must be present for avoid biased results), but also a visit to the human medical center (ICAMEN) was needed. Therefore, we had to complete this study for a long period to obtain enough data. Together with the exclusion criteria, the number of pairs animals/owners is limited. Then we must complete the study for a two-year period to obtain enough animal/owners for its statistical analysis. Then, the 198 pairs were obtained in this period.

Another reviewer has recommended to change the last sentence on the introduction to another place within the manuscript. We have moved to the Material and Method and a new sentence has been added for clarifying the readers the number of animals/owners included in the study.

To obtain enough data for an appropriate statistical analysis it was needed to complete this study for a two-year period.

For the p-values in Table 2, it will be helpful for the reader if you clarify what test was used to obtain that p-value.

A clarification for readers has been included in the title of the Table:

Table 2. Results of the studied variables related to the characteristics of the owners. The distribution of categorical variables was estimated using the 2 test and, when required, Fisher's exact test was used.  Statistical differences are highlighted*

Line 419 - I am not a veterinarian, but please check to determine the appropriateness of a veterinarian recommending physical activity to pet owners "for the benefit of both." I assume that it is not appropriate for vets to make recommendations regarding human health.

This study has been completed from a One Health approach, then we have human physician in our research team. We appreciate your comments and has included a modification in the sentence to clarify this to the readers. Thanks!

 We suggest that veterinarians and physicians from a One Health approach

must develop and design strategies to encourage pet owners to engage in physical activity with their dogs for the benefit of both.

Minor Comments:

  • Line 215 - Only the proper names (Yorkshire, German, Canarian, etc.) should be capitalized. Bulldogs/Boxers/Poodles should not be capitalized.
  • The number of decimal places (one or two) is not consistent when presenting percentages. I suggest picking one or the other for consistency. The same goes for decimal places for confidence intervals (ex. Line 303).
  • Line 315 - use the abbreviation only here (BMI had already been defined). The same goes for APOP in Line 337. (Also, there are two periods in this sentence).
  • Line 339 - clarify that you mean especially if the women (not the dogs) do not exercise, if this is your intent.
  • Line 416 - overweight dogs were not more prevalent in overweight women - obesity in dogs was more prevalent when their owners were overweight women older than 40 years with little education who did not engage in physical activity.

Thanks, all these comments have been considered.

Reviewer 3 Report

This is an interesting study to add to the discussion of the problem and interconnection between overweight dogs and their owners.  It adds to the previously published works, and is interesting as there is a higher incidence of unaltered dogs in comparison to studies, for example, in the United States.

The relationship between overweight dogs and their owners is well known at least anecdotally to primary care veterinarians.   It is a delicate issue to address without offending the owners.  It is not as easy as the authors suggest to design a weight and exercise program and have the owners accept and implement it.  

I believe it will be interesting to readers and will encourage further studies.  

Author Response

Thank you very much for your review. Your suggestions are greatly appreciated. 

A thorough English revision has been performed. In fact, we have sent the manuscript to a native English speaker and professional English proofreader. That is why some errors have been detected and some improvements in English expression have been incorporated into the text. We believe that the text has now clearly improved thanks also to the input of the reviewers. 

Once again, we thank you for your contribution to the improvement of the manuscript.

Round 2

Reviewer 1 Report

Authors aimed to describe the influence of dog ownership on canine obesity and bring a new perspective for strategies to reduce pet obesity and pet health through a “One Health” approach in a in a very interesting way.

Authors made the inclusions, suggestions and modifications indicated by the reviewers in a satisfactory manner. Significantly improving the quality and understanding of this new version of the manuscript.

No new comments arise by this reviewer

This manuscript is a resubmission of an earlier submission. The following is a list of the peer review reports and author responses from that submission.

Round 1

Reviewer 1 Report

Authors aims to evaluate factors related to age, gender, weight status, lifestyle and physical 87 activity of owners which contribute to and influence canine obesity.

Specific comments/suggestions are highlighted in the attached .pdf file

This is a very interesting topic. I think that M&M section should include more details on the data analysis, especially because it is this that defines the results that are being observed. Check the comments about tables.

Reviewer 2 Report

This manuscript describes a study of 198 pairs of owners and dogs that seeks to identify which characteristics of the owners are associated with excess weight in dogs. This is not a new idea (many studies have examined how owners’ beliefs and behaviour are associated with, and predict, the weight of companion animals), but the present work is purportedly novel in focusing on owners in the Canary Islands, which is apparently a region where people are often obese.

There are some strengths to the work (e.g., that all dogs underwent a clinical examination, including assessment of BCS by a veterinarian), but overall, I don’t believe that the present findings contribute to understanding for the reasons below:

  1. Recruiting owners from an area where people are likely obese does not seem like a significant contribution (e.g., other studies have also included overweight owners), and could even be a limitation in the sense that it restricts the range of human weight and, thus, the assessment of it’s association with the weight of the companion animals.
  2. The work is limited by a focus on the demographic and behavioural characteristics of owners, rather than their beliefs or other factors underlying these behaviours. As such, it offers little explanation as to why these characteristics are associated with excess weight among dogs (e.g., why are women over 40 years of age more likely to have overweight dogs?). Indeed, the authors have to speculate that their results might indicate “that owners may apply their personal attitudes and behaviour to their pets”, when this could have been empirically tested. This limitation also means that the research does not suggest potential modifiable targets for intervention (we cannot seek to reduce obesity by changing the gender of owners or making them younger).
  3. One association that is potentially more useful is with owners’ level of physical activity (i.e., owners who are not physically active themselves are more likely to have overweight dogs), but this is not a novel insight – and indeed, experimental work has already built on this finding (e.g., the People and Pets Exercising Together project, Kushner et al., 2006).

In short, I don’t believe that the present findings contribute to understanding.

More minor issues

  1. Why dichotomise BCS into overweight vs. normal weight, when the scale is continuous? Treating the scale as intended, would allow the authors to differentiate overweight from obese dogs, as well as use multiple linear regression, rather than logistic regression.
  2. Similarly, the measures of owner’s behaviour (e.g., alcohol consumption and physical activity) are very crude (i.e., dichotomised), which is especially unfortunate given the significant work that has gone into developing self-report measures of these behaviours., some of which are relatively brief.
  3. The use of language needs some attention, not least to avoid awkward nominalisations like “a human high obesity prevalence region” (e.g., rephrase to “A region with high levels of human obesity”)

Reviewer 3 Report

This study deal with the owners’ characteristics (e.g., lifestyle, weight) that could influence canine obesity. To this aim, the authors use a One Health approach.

The paper is well written and the writing is smooth and simple. Overall, the paper is interesting.

I have only some minor revisions (see below).

Line 42: specified the meant of BCS in parenthesis.

Line 193: add the commas after “weight” and before “were”.

Line 198: add the spaces before and after “=”.

Line 198: add the comma after “However”.

Line 230: It refers to “human” or “man”?

Line 234: remove “(OR = 3.054; P = 0.001)” from the discussion section; if necessary, it can be added in the results section (line 205-6).

Lines 251 and 254: add the spaces before and after “=”.

Line 257: replace “Tables” with “tables”.

Line 284: add the spaces before and after “=”.

Reviewer 4 Report

Major comments:

  • The statistical comparison is not at all clear in the Tables. What is being compared to what to generate the p value in, for example, Table 2, BMI, p value <0.001? Presumably the intent is that overweight owners have overweight dogs but it needs to be made clear. The 'Total' column is also not immediately intuitive as to its purpose. Explain why it matters if we know how many owners are overweight/obese vs not. If it matters, that p value should be calculated and listed, too.
  • The conclusion's first paragraph is good and related to the manuscript's data. The remainder is essentially reiterating the current literature and should be focused on recommendations based on the data in the manuscript. Also, if the purpose of the manuscript is to prevent and treat obesity, and all those factors were examined, shouldn't the recommendations address those? Granted, it will be hard to state that veterinarians should counsel overweight, uneducated women over 40 differently about their pet's obesity, but to not address it at all weakens the manuscript.
  • The disparity in group sizes should be addressed and perhaps justified-- why were 137 needed in the overweight group if only 61 were used in the normal weight group? 

Minor comments:

  • Line 29 - Suggestion to summarize what the statement says
  • 34: varies between → ranges from 25 to 44%
  • 37: remove one of the % symbols
  • 38: typically overweight rather than over-weight
  • 40-42: First - define BCS and second - I don't understand what this sentence is saying. How does this phrase enable veterinarians to use other strategies? Update - I see that BCS is defined line 111. Move the definition to 40-42.
  • 43: This paragraph seems out of place
  • 49: influences rather than influence
  • 60: This needs to be introduced differently; it seems out of place after the previous paragraph talks about exercise -- there are no data on obesity in that paragraph; in fact the paragraphs starting with line 48 and 56 could be kept together and moved elsewhere. They are out of place with the rest of the narrative.
  • 69: this paragraph is similar enough to the paragraph starting with line 43. could be removed perhaps or expanded if not
  • 76: this has already been hinted at when cultural differences in exercising with pets were mentioned. Recommend rephrasing
  • 78: define the relationship. It could be an inverse relationship (although I know that's not likely)
  • 84: the word however seems out of place here
  • 85: Define the One Health Approach. I'm not familiar with that
  • 110: odd that the new standards were introduced in the introduction (38-41) if they aren't going to be used here. Perhaps expand on why you chose BCS rather than the new standards.
  • 115: language is a bit confusing. Perhaps: '...overweight group comprised 137 dogs with excess weight (overweight and obese) paired with their owners'; same language for the control group
  • 121: I feel certain it was, but it's not clear whether the human part of the study protocol underwent ethical review. Perhaps reiterate the ethical approval from the Animals section or move the ethical approval to its own section
  • 126: '(referring to the owners)' should be removed, unless you want to put it at the end of the list of characteristics. Otherwise we are made to wonder whether the dogs' employment situations were measured, for example
  • 131: because these groups were combined for statistical analysis, did they exist as separate groups in any other way? If not, say that. 
  • 133: BMI has already been defined. Not necessary here
  • 161: 'entire' is not a term I'm familiar with in this context. Does that mean the same as 'intact' in the States? 
  • Table 2: BMI (>25 kg/m2)---- this is confusing because that value refers to only overweight/obese and not to both in the category --- suggestion for just BMI and then after overweight/obese state BMI >25 kg/m2 and after normoweight BMI less than or equal to 25
  • 240 - why is the identification by dog owners concerning? is it the lack of owners' identifying their animals as overweight? or is that they identify them as overweight but yet consider them healthy? recommend rephrasing for clarity
  • Sentence started at the end of line 245 through the end of the paragraph needs references
  • 247- protential → potential
  • 254: probably OR rather than AND
  • 255-256: recommend consistency with the number of decimal points when comparing percentages -- personally think that two decimal points is overkill
  • 267: BCS has already been defined; no need to define again
  • 269: not enough to state that the data contrast; speculate as to the reason
  • 277: use the abbreviation defined earlier: BMI
  • 278: 'perception of people about obesity and overweight of their pets' is not clear; rephrase for clarity
  • paragraph starting with 281 - the first topic dealing with age is good and needed. The remaining sentences seem out of place at best and repeated at worst.
  • 293-294: again, need consistency with number of decimal places in percentages
  • 294: again it seems repeated to state again about no physical exercise and women
  • 295: the last two sentences of the paragraph are out of place and should be moved or removed
  • Sentence starting on 312 to the end of the paragraph -- seems more suited for the conclusion than this paragraph